# Synergistic Adsorption-Catalytic Sites TiN/Ta_2_O_5_ with Multidimensional Carbon Structure to Enable High-Performance Li-S Batteries

**DOI:** 10.3390/nano11112882

**Published:** 2021-10-28

**Authors:** Chong Wang, Jian-Hao Lu, Zi-Long Wang, An-Bang Wang, Hao Zhang, Wei-Kun Wang, Zhao-Qing Jin, Li-Zhen Fan

**Affiliations:** 1Beijing Advanced Innovation Center for Materials Genome Engineering, Institute of Advanced Materials and Technology, University of Science and Technology Beijing, Beijing 100083, China; wangchong18810@163.com; 2Military Power Sources Research and Development Center, Research Institute of Chemical Defense, Beijing 100191, China; hahalujianhao@163.com (J.-H.L.); wangzilong0709@163.com (Z.-L.W.); wab_wang2000@163.com (A.-B.W.); dr.h.zhang@hotmail.com (H.Z.)

**Keywords:** lithium-sulfur batteries, catalyst, TiN/Ta_2_O_5_, multidimensional carbon

## Abstract

Lithium-sulfur (Li-S) batteries are deemed to be one of the most optimal solutions for the next generation of high-energy-density and low-cost energy storage systems. However, the low volumetric energy density and short cycle life are a bottleneck for their commercial application. To achieve high energy density for lithium-sulfur batteries, the concept of synergistic adsorptive–catalytic sites is proposed. Base on this concept, the TiN@C/S/Ta_2_O_5_ sulfur electrode with about 90 wt% sulfur content is prepared. TiN contributes its high intrinsic electron conductivity to improve the redox reaction of polysulfides, while Ta_2_O_5_ provides strong adsorption capability toward lithium polysulfides (LiPSs). Moreover, the multidimensional carbon structure facilitates the infiltration of electrolytes and the motion of ions and electrons throughout the framework. As a result, the coin Li-S cells with TiN@C/S/Ta_2_O_5_ cathode exhibit superior cycle stability with a decent capacity retention of 56.1% over 300 cycles and low capacity fading rate of 0.192% per cycle at 0.5 C. Furthermore, the pouch cells at sulfur loading of 5.3 mg cm^−2^ deliver a high areal capacity of 5.8 mAh cm^−2^ at low electrolyte/sulfur ratio (E/S, 3.3 μL mg^−1^), implying a high sulfur utilization even under high sulfur loading and lean electrolyte operation.

## 1. Introduction

Lithium-sulfur (Li-S) batteries are ideal candidates to substitute conventional lithium-ion batteries [1]. However, despite their attractive merits including high theoretical energy density and abundant resources, the practical uses of Li-S batteries are still hampered by a series of deleterious defects facing widespread commercialization, such as the intrinsic poor electrical conductivity of sulfur and discharge products (Li_2_S/Li_2_S_2_), the shuttling phenomenon originated from the dissolution of lithium polysulfide (LiPSs), and, particularly, the sluggish conversion of LiPSs to solid lithium sulfides, bringing about low sulfur utilization, fast capacity fading, and poor cyclability [2].

Accordingly, strategies have been developed to solve the above problems in the past few decades. It has been corroborated that the conversion of LiPSs on the interface between the electrolyte and host materials is decided by their moderate interactions and fast electron/ion exchange [3]. Catalysts such as transition metal-free polar materials [4,5], transition metal compounds [6,7,8,9], and metals [10] can not only capture LiPSs to decrease their dissolution and diffusion in the electrolyte but also boost the conversion between LiPSs and Li_2_S_2_/Li_2_S. Among them, titanium nitride has been widely used as the catalytic material [11,12], due to its high electrical conductivity, which can propel the kinetics of the transformation of LiPSs. Goodenough and coworkers reported a mesoporous TiN with a high surface area, benefiting from its intrinsic electrical conductivity, fine porous framework, and appropriate adsorption ability of TiN; the TiN-S composite cathode exhibits high specific capacity and excellent rate capability [13]. However, TiN nanoparticles (TiN NPs) cannot effectively suppress the shuttle effects of LiPSs, because of their weak adsorption capacity toward LiPSs [14]. Kim et al. proposed an effective electrocatalyst Ta_2_O_5_ for LiPSs conversion in the Li-S system; the intrinsic chemical polarity of Ta_2_O_5_ could establish favorable interactions with the LiPSs [15]. However, Ta_2_O_5_ demonstrates insufficient electrical conductivity, because of its electron band structure [16], which does not provide desirable electron mobility and catalytic activity. Therefore, it is difficult to make full use of catalysis ability depending on the sole component, which is short in either adsorption or electrical conductivity. Especially with high sulfur loading for a thick cathode, the sluggish and incomplete conversion of LiPSs to solid lithium sulfides limits the full utilization of intermediates, leading to the shuttle effect and rapid capacity decay during cycling.

To attain high specific capacity under the condition of high sulfur content of the sulfur-based composite and thick sulfur cathode, we designed the TiN@C/S/Ta_2_O_5_ cathode for Li-S batteries. Both TiN and Ta_2_O_5_ have a synergy enhancement effect to promote the affinity with LiPSs and speed up the kinetics of sulfur conversion reaction. In addition, the multidimensional carbon structure, which is the mixture of Super P, CNT, and graphene, can offer high electrical conductivity and sustain the strain generated by the volumetric changes of the active materials during cycling [17]. Their characteristic superiorities endow our high fraction of sulfur cathode with good rate response capability and superior cyclability even under high sulfur loading and lean electrolyte/sulfur ration operation.

## 2. Experimental Section

### 2.1. Preparation of TiN@C/S/Ta_2_O_5_ Composite

The TiN@C/S/Ta_2_O_5_ composite was fabricated in a typical liquid-phase suspension process. A certain amount of Super P, CNT, graphene, and TiN NPs (the weight ratio of Super P: CNT: G: TiN = 2:2:1:10) was ball-milled to obtain the uniform slurry. Sulfur was synthesized based on the reaction between Na_2_S_2_O_3_ and HCOOH [18]; the suspension of sulfur was injected into the mixed solution of TiN@C under vigorous stirring for more than 10 h. After that, the sediment was obtained by filtering, washed with distilled water several times to wipe off the soluble impurities, and dried under vacuum 60 °C for 24 h. Next, the TiN@C/S materials were uniformly dispersed in distilled water again, and an appropriate amount of Ta(OEt)_5_ was added to the above suspension; the amorphous Ta_2_O_5_ was produced by the hydrolysis reactions between Ta(OEt)_5_ and deionized (DI) water [19]. Subsequently, the mixture was stirred all night. Finally, the TiN@C/S/Ta_2_O_5_ (the weight ratio of Super P: CNT: G: TiN: Ta_2_O_5_ = 2:2:1:10:10) composite was collected by centrifugation, washed with deionized water several times, and dried at 60 °C for 24 h. The TiN@C/S (SuperP: CNT: G: TiN = 2:2:1:20) and Ta_2_O_5_@C/S (Super P: CNT: G: Ta_2_O_5_ = 2:2:1:20) composites were prepared with the procedure similar to TiN@C/S/Ta_2_O_5_ composite; the content of sulfur in both TiN@C/S and Ta_2_O_5_@C/S materials is also 90 wt%.

The sulfur composite cathodes were prepared by mixing the active material (TiN@C/S/Ta_2_O_5_), Super P, carbon nanotube, and a binder (LA133) in deionized water and isopropanol mixed solution with a weight ratio of 80:5:5:10. After stirring for 12 h, the cathode slurry was blade-cast onto Al foils, followed by drying at 60 °C for 24 h. Similarly, the TiN@C/S and Ta_2_O_5_@C/S composites were prepared. 

### 2.2. Material Characterization Techniques

Thermogravimetric analysis (NETZSCH TG 209F3, NETZSCH Gerätebau GmbH, Selb, Germany) was carried out with a heating rate of 5 °C min^−1^ under an atmosphere of N_2_. The metallic element content was measured by ICP-OES (Agilent 725ES & Agilent 5110, Agilent Technologies, Santa Clara, CA, USA). The morphology and structure of materials were recorded using scanning electron microscopy (Zeiss G300, Carl Zeiss Inc., Thornwood, New York, NY, USA) and transmission electron microscopy (JE0L Ltd., Tokyo, Japan). The elemental compositions and crystal structures of these samples were analyzed by X-ray photoelectron spectroscopy (XPS) (Thermo Scientific K-Alpha, Thermo Fisher Scientific, Waltham, MA, USA) and X-ray diffraction (Ultima IV, Rigaku Corporation, Tokyo, Japan).

### 2.3. Polysulfides Adsorption Experiment

Li_2_S_6_ solution with a concentration of 10 mmol L^−1^ was prepared by mixing lithium sulfide (Li_2_S) and sulfur power with a molar ratio of 1:5; the mixture was added into 1,3-dioxolane (DOL)/1,2-dimethoxyethane (DME) (1:1, *v*/*v*) solution, followed by intense stirring for 24 h in an Ar atmosphere. A total of 20 mg of TiN, Ta_2_O_5_, and TiN/Ta_2_O_5_ were added into 30 mL of Li_2_S_6_ solution, respectively, and rested for 12 h. The supernatant and precipitates were researched by UV-vis spectrophotometry and XPS.

### 2.4. Assembly of the Symmetric Cell 

The electrode powers (TiN, Ta_2_O_5_, and TiN/Ta_2_O_5,_), CNT, and polyvinylidene fluoride (PVDF) binder were dispersed into NMP with a weight ratio of 70:20:10 to form a homogeneous solution, and then it was coated onto the current collector. The symmetric cell used the electrodes as both cathode and anode, 30 µL electrolyte containing 0.5 mol L^−1^ of Li_2_S_6_, and 1.0 mol L^−1^ of LiTFSI in a 1:1 (*v*/*v*) mixture of 1,3-dioxolane (DOL), and 1,2-dimethoxyethane (DME) was added into each coin cell. Cyclic voltammetry (CV) and electrochemical impedance spectroscopy (EIS) measurements were collected on an electrochemical workstation (VersaSTAT3, ametek, Berwyn, PA, USA).

### 2.5. Electrochemical Measurement

Standard CR2025 coin cells were assembled using the prepared electrodes and polypropylene separator (Celgard 2400, Celgard, Charlotte, NC, USA), with lithium metal as the anode. Charge-discharge performances of both coin cells and pouch cells were tested between 1.8 and 2.6 V using a LAND CT2001A (Landian, Wuhan, China) multi-channel battery testing system at room temperature. 

In this experiment, the electrolyte was 1M LiTFSI in DOl/DME (1:1 *v*/*v*) containing LiNO_3_ as an additive (1 wt%), The E/S ratio in the coin cells with areal sulfur loading (1.5 mg cm^−2^) was controlled to be 10 μL mg^−1^. The pouch cells have average areal sulfur loading of 5.3 mg cm^−2^ and a decreased electrolyte/sulfur ratio of 3.3 μL mg^−1^. 

## 3. Results and Discussion

The fabrication process of TiN@C/S/Ta_2_O_5_ composite is shown in Figure 1. Firstly, Super P, CNT, graphene, and TiN were mixed and dispersed in deionized water to obtain the homogenous host materials. Sulfur was synthesized based on a disproportionated reaction. Then, the suspension of sulfur nanoparticles was added into the above host materials system. Through long-time stirring, all those materials were dispersed homogeneously without agglomeration, and the sulfur nanoparticles were evenly deposited in the TiN@C host. Finally, Ta(OEt)_5_ was added to TiN@C/S slurry to shape the amorphous Ta_2_O_5_ by the hydrolysis reactions, which were well-dispersed on the external surface of TiN@C/S/Ta_2_O_5_.

Based on thermogravimetric analysis (TGA), the rationale design TiN@C/S/Ta_2_O_5_ composite displays a sulfur content of 90 wt% in the sulfur composite (Figure 2a). Therefore, it is difficult for the low content 5% of both TiN and Ta_2_O_5_ to respond in X-raydiffractionpatterns (XRD) measurement (Ultima IV, Rigaku Corporation, Tokyo, Japan). The characteristic peaks of the TiN@C/S/Ta_2_O_5_ composites are following the standard sulfur PDF card S (JCPDS 08-0247) (Appendix A), while the diffraction peaks of TiN and Ta_2_O_5_ can scarcely be found. To prove the existence of TiN and Ta_2_O_5_, the TiN@C/S/Ta_2_O_5_ composites were washed with carbon disulfide to wipe off redundant sulfur. TiN diffraction peaks, which are in accordance with the standard PDF card of TiN, and amorphous tantalum oxide are discovered in the rinsed sample (Figure 2b). The mass ratio of Ta and Ti in TiN@C/S/Ta_2_O_5_ are 3.78 wt% and 3.52 wt% by ICP-OES testing. The surface chemical states of TiN@C/S/Ta_2_O_5_ were investigated by XPS under high vacuum, the display of strong peak intensity of Ta, and weak intensity Ti, further revealing the existence of an out-coated Ta_2_O_5_ layer in Figure 2c,d. The pore structures of the multidimensional carbon structure were studied using N_2_ adsorption-desorption analysis. The BET-specified surface area of the multidimensional carbon structure was 352 m^2^ g^−1^. A dual distribution of micropore and mesopore was observed on the multidimensional carbon structure (Figure 2e,f); its electronic conductivity was 4.38 × 10^3^ S m^−1^.

The morphology of the TiN@C/S/Ta_2_O_5_ composites was investigated by scanning electron microscope (SEM); the TiN@C/S/Ta_2_O_5_ composite consists of a pile of clusters with an average size of about 0.2–1 μm (Figure 3a). The rough surface and hollow morphology guarantee the intimate contact between sulfur composites and electrolytes, leading to fast ion and electron transportation (Figure 3b).

The end products of the TiN@C/S/Ta_2_O_5_ composite electrode were displayed by transition electron microscope (TEM) observation. The closely packed nano-clusters structure is further conformed (Figure 4a). As displayed in the high-magnification TEM image (Figure 4b), the lattice spacings at 0.212 nm correspond to the (200) plane of TiN. Meanwhile, as shown in the (EDS) elemental mapping images in Figure 4c,g, Ti, N, Ta, O, and S elements are observed; these results manifest the formation of TiN@C/S/Ta_2_O_5_ composite. In addition, the EDS elemental mapping images of one single bulk TiN@C/S/Ta_2_O_5_ show that more signals of Ta and O can be observed, and most of them are on the surface of the unit; this result confirms the formation of the external Ta_2_O_5_ coating layer. Moreover, the existence of a few elements Ti and N imply that most TiN may be implanted inside the TiN@C/S/Ta_2_O_5_ composite (Appendix A).

Polysulfides adsorption experiments were systematically implemented using the Li_2_S_6_ electrolyte for modeling polysulfide intermediates. In contrast, the Li_2_S_6_ solution turns almost transparent after being adsorbed by Ta_2_O_5_ and TiN/Ta_2_O_5_, while an obvious yellowish color still can be viewed for that with TiN. The absorbance was analyzed by UV-vis absorption spectra; it can be seen that the adsorption intensity with Ta_2_O_5_ and TiN/Ta_2_O_5_ solution decreases compared with others (Figure 5a). The X-ray photoelectron spectroscopy (XPS) technique was put into use to understand the chemical interaction of catalysts before and after the absorption of Li_2_S_6_; it can be observed that Ti 2p peaks have hardly any binding energy shift before and after the adsorption of Li_2_S_6_ (Figure 5b). However, the additional peak at 407.3 and 399.3 eV in the N 1 s spectrum of the Li_2_S_6_-treated TiN corresponds to Ti-N-S bonding (Figure 5c), which should be ascribed to the electronegativity and polar of nitrogen; it ensures a strongly interaction with LiPSs and means that the exposed N sites are utilized as the main active sites for absorbing Li_2_S_6_ [20]. After interacting with Li_2_S_6_, a large positive shift can be observed in both Ta 4f and O1s peaks in (Figure 5d,e); the shifts of these peak positions are considered to be produced by the strong binding interaction between Li_2_S_6_ and Ta_2_O_5_, confirming their strong LiPSs adsorption capability [15].

Symmetrical cells were assembled using TiN, Ta_2_O_5_, and TiN/Ta_2_O_5_ identical electrodes and Li_2_S_6_ electrolyte to investigate the LiPSs conversion dynamics [21]. The TiN/Ta_2_O_5_ compound presents the strongest redox current peaks among different samples (Figure 6a), which could be attributed to the integrated adsorption and catalytic ability of the TiN/Ta_2_O_5_ compound. Ta_2_O_5_ shows strong adsorption capability for polysulfides, but their intrinsically low electrical conductivity will impede the reaction kinetics of soluble LiPSs conversion into insoluble Li_2_S/Li_2_S_2_. Similarly, although TiN exhibits good electrical conductivity, its weak affinities with lithium polysulfides cannot retain LiPSs to suppress the shuttling effect. Moreover, electrochemical impedance spectroscopy (EIS) analysis was employed to understand the interfacial charge transfer kinetics [22] (Figure 6b). TiN/Ta_2_O_5_ shows the smallest charge transfer resistance; the interface impedance of symmetric cells can explain the chemical affinity ability of the electrode materials to LiPSs and the ability of TiN/Ta_2_O_5_ to accept electrons when interacting with LiPSs, further reflecting its fast charge transfer and facile sulfur redox reactions at the TiN/Ta_2_O_5_ and polysulfides interface. All the above results reveal that the synergistic adsorptive-catalytic effect of TiN/Ta_2_O_5_ toward enhanced LiPSs conversions.

To investigate the improved reaction kinetics of the TiN@C/S/Ta_2_O_5_ composites, the expedited polysulfides redox kinetics between solid-liquid-solid conversions was further indicated by the CV curves of the cells with TiN@C/S/Ta_2_O_5_, Ta_2_O_5_@C/S, and TiN@C/S electrodes at a scan rate of 0.1 mV s^−1^ [23]. All the assembled cells show the typical pair of redox peaks. Compared with Ta_2_O_5_@C/S and TiN@C/S, CV curves of the TiN@C/S/Ta_2_O_5_ electrode display visibly stronger peak current intensity and closer peak position (Figure 7a). Additionally, the CV curves overlap well upon several cycles (Appendix A). At the same time, the voltage plateaus of galvanostatic charge/discharge profiles are also correspondent to peaks in the CV curves. The TiN@C/S/Ta_2_O_5_ cathode exhibits the highest specific capacity and a relatively lowest overpotential at 0.5C during the first cycle [24] (Figure 7b), when adsorptive-catalytic sites, both TiN and Ta_2_O_5_, were implanted within the multidimensional carbon structure. On the one hand, the dispersed-distribution Ta_2_O_5_ coating suggests stronger LiPSs adsorption capability due to the polar Ta-O bonding; it is found that long-chain polysulfides Li_2_S_6_ and Li_2_S_8_ are easily deformed after adsorbing on the surface of strong polar active sites Ta_2_O_5_. In this way, the Ta_2_O_5_ facilitates the fragmentation reactions of long-chain polysulfides into shorter chains and accelerates the kinetics of polysulfides conversion reactions by reducing the activation energy [25]. On the other hand, its high electrical conductivity of TiN can improve the reduction reaction kinetics by promoting electron transport in the electrode, resulting in rapid conversion of LiPSs into Li_2_S. The TiN@C/S/Ta_2_O_5_ electrode owns the dual advantages of excellent trapping capability of LiPSs (Ta_2_O_5_) and superior electronic conductivity (TiN) to achieve the adsorption-catalysis synergy. The rate properties and corresponding charge-discharge profiles of these electrodes are displayed (Figure 7c and Appendix A); the TiN@C/S/Ta_2_O_5_ delivers the best rate performance with the highest discharge capacity of 1112 mAh g^−1^ at 1C compared with Ta_2_O_5_@C/S (942 mAh g^−1^) and TiN@C/S (993 mAh g^−1^), the reversible capacity of 1216 mAh g^−1^ when the current returns to 0.1 C. These results furtherly confirm the improved catalytic activity and kinetics for LiPSs conversion, which is due to the elaborate design of TiN@C/S/Ta_2_O_5_ composite. The charge/discharge voltage profiles of TiN@C/S/Ta_2_O_5_ at various current densities are illustrated (Appendix A), TiN@C/S/Ta_2_O_5_ is capable of retaining the two-plateau discharge profile at a raised rate of up to 1 C without the severe electrochemical polarization-induced serious deformations of the voltage profile. The long-term cyclability performances are compared at a high current density of 0.5 C; the TiN@C/S/Ta_2_O_5_ cathode exhibits the optimal performances with an initial discharge capacity of 1175 mAh g^−1^ at 0.5 C, which is much higher than all the other cathodes. The TiN@C/S/Ta_2_O_5_ cathode still delivers a decent discharge capacity of 660 mAh g^−1^ and high Coulombic efficiency after 300 cycles (Figure 7d). The gentle capacity attenuation and prominent capacity retention of TiN@C/S/Ta_2_O_5_ cathode benefit from its abundant polysulfide-trapping and catalytic active sites: the out-coated Ta_2_O_5_ can serve as a covering layer to physically restrain part of LiPSs inside and chemically adsorb another part of out-diffused LiPSs on its polar active surface. Furthermore, TiN NPs presents satisfied catalysis ability to catalyze the conversion of LiPSs, originating from expediting electron transfer. With the synergistic and complementary roles of the cathode materials, the TiN@C/S/Ta_2_O_5_ cathode improves the efficient utilization of lithium polysulfides and promotes the chemical interaction with LiPSs and sulfur redox kinetics. It is worth noting that a distinct two-plateau discharge profile maintains well from the 1st to the 100 th cycle at a relatively high rate of 0.5 C (Appendix A), which is consistent with the results of cycling performance of TiN@C/S/Ta_2_O_5_ over 300 cycles at 0.5 C. This result is very competitive compared with the previously reported other electrodes (Appendix A) [26,27,28] but it is under the condition of high S content up to 90 wt% and relatively low content of catalysts and carbon materials. 

The wonderful performance of the TiN@C/S/Ta_2_O_5_ coin cell inspired us to fabricate a pouch cell with 200 mg sulfur loading in a single-piece cathode with dimensions of 75 mm × 50 mm. The pouch cell was cycled at a current density of 200 mA g^−1^, the pouch cell showed a specific capacity of over 1100 mAh g^−1^ with a capacity retention rate of 61.63% for 50 cycles (Figure 7e). Even the uniform Li metal corrosion caused by the dissolution and diffusion of LiPSs in the electrolyte could result in the fluctuation of Coulombic efficiency with limited Li, but the pouch cell still shows high Coulombic efficiency of ˃96% and stable cycle life, further suggesting the function of the outer layer Ta_2_O_5_ in restraining the dissolution of LiPSs in organic electrolyte and eliminating Li metal corrosion. Our TiN@C/S/Ta_2_O_5_ pouch cell exhibits a superior electrocatalytic sulfur reduction reaction (SRR) and represents a significant advance in the light of specific capacity, good cycling life, and capacity retention when compared with some reported data (Appendix A) [29,30,31,32]. Despite the substantial progress made in our work, there is much room to further enhance the energy densities by optimizing both the mass-production process and cell configuration for making the pouch cell.

## 4. Conclusions

In summary, the concept of combining the merits of rational materials to construct the high-energy-density lithium-sulfur battery has been introduced. The TiN@C/S/Ta_2_O_5_ composites with high sulfur fraction were synthesized via the co-precipitation method through a simple and low-cost preparation process. The novel design of TiN@C/S/Ta_2_O_5_ has demonstrated excellent cyclability and rate capability, owing to its potential for inhibiting the shuttle effect and facilitating LiPSs redox reaction in LSBs. Our work will guide the widespread commercialization of high-energy-density Li-S batteries.

## Figures and Tables

**Figure 1 nanomaterials-11-02882-f001:**
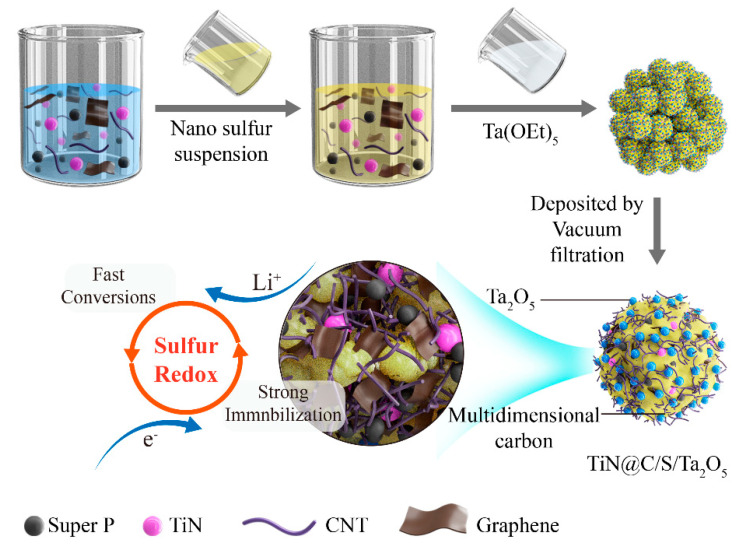
Schematic illustration of the synthesis route of the TiN@C/S/Ta_2_O_5_ composites.

**Figure 2 nanomaterials-11-02882-f002:**
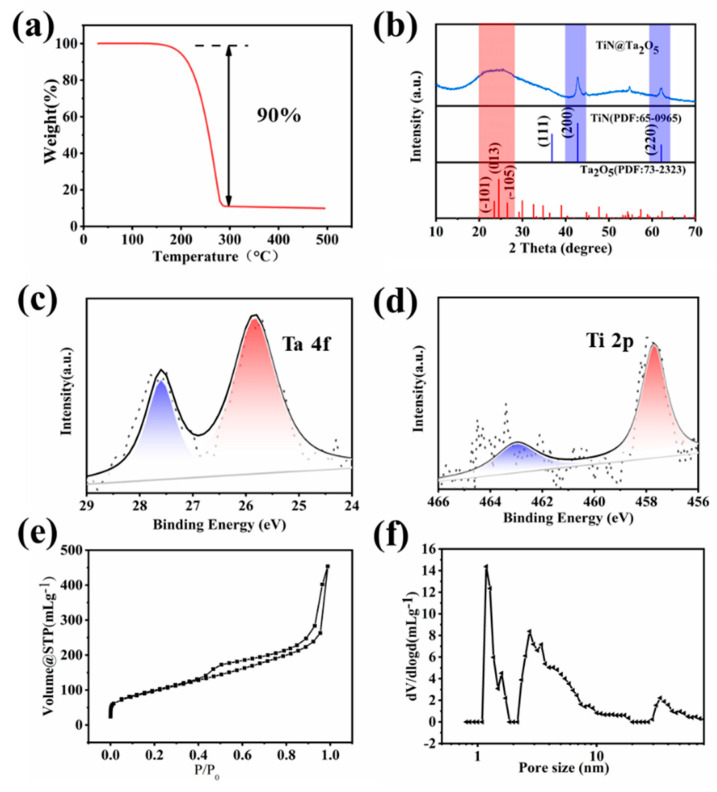
(**a**) TGA of TiN@C/S/Ta_2_O_5_ composites. (**b**) XRD patterns of TiN@C/Ta_2_O_5_ composites. (**c**,**d**) XPS spectra of TiN@C/S/Ta_2_O_5_ composites. (**e**) N_2_ sorption isotherm and (**f**) pore size distribution based on QSDFT model of the multidimensional carbon structure.

**Figure 3 nanomaterials-11-02882-f003:**
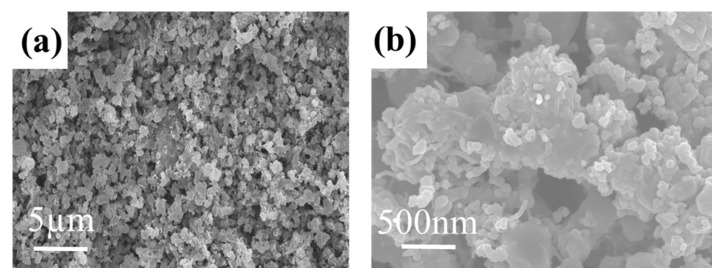
(**a**,**b**) SEM image of TiN@C/S/Ta_2_O_5_ composite.

**Figure 4 nanomaterials-11-02882-f004:**
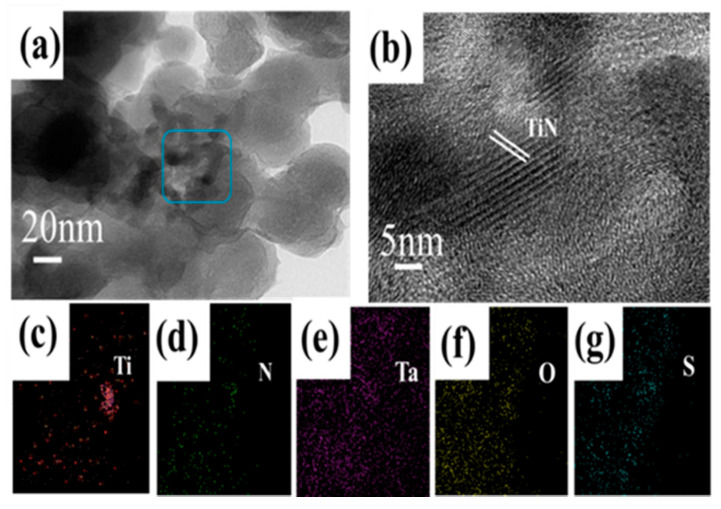
(**a**) TEM image of TiN@C/S/Ta_2_O_5_ material. (**b**) High-resolution TEM image of TiN@C/S/Ta_2_O_5_ material. (**c**–**g**) EDS elemental mapping images of TiN@C/S/Ta_2_O_5_ material with the selective regions shown in (**a**).

**Figure 5 nanomaterials-11-02882-f005:**
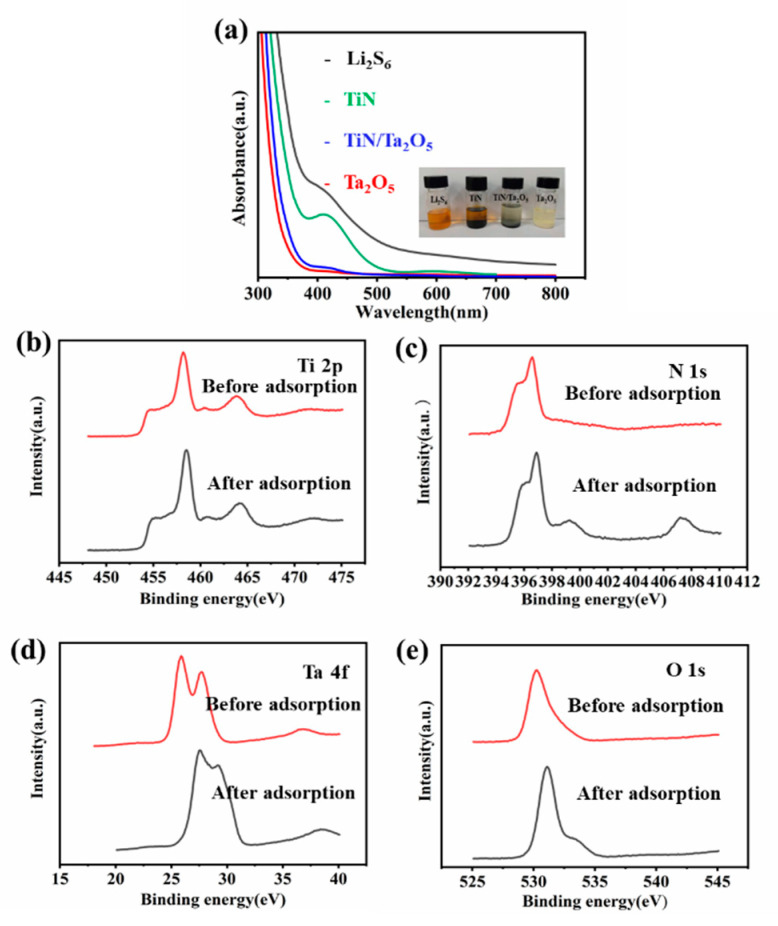
(**a**) UV-vis spectra of the Li_2_S_6_ solution with TiN, Ta_2_O_5_, TiN/Ta_2_O_5_, and bare Li_2_S_6_ solution. (**b**–**e**) XPS spectra of Ti 2p, N 1s, Ta 4f, and O1s before and after adsorbed Li_2_S_6_.

**Figure 6 nanomaterials-11-02882-f006:**
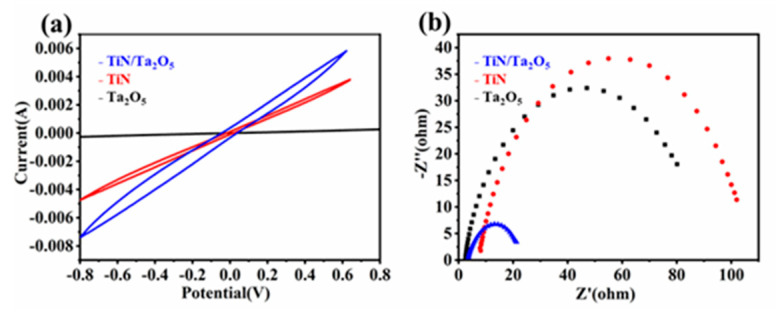
(**a**) CV curves of TiN, Ta_2_O_5_, and TiN/Ta_2_O_5_ symmetric cells with 0.1 M Li_2_S_6_ electrolyte. (**b**) EIS spectra of Li_2_S_6_ symmetrical cells.

**Figure 7 nanomaterials-11-02882-f007:**
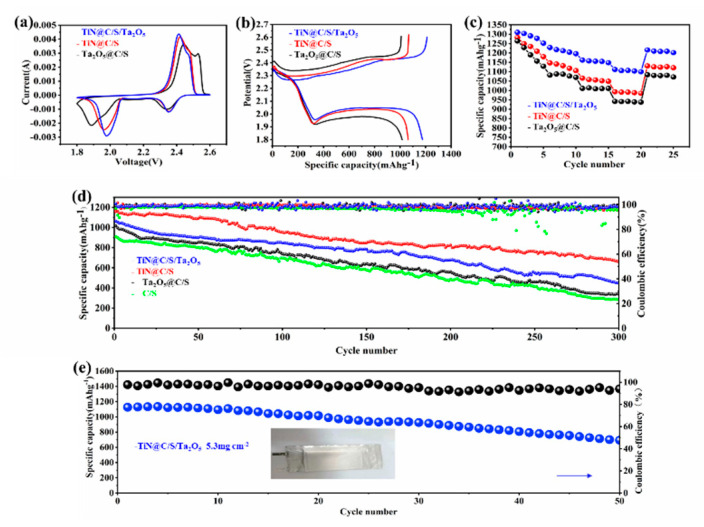
(**a**) CV curves of TiN@C/S/Ta_2_O_5_, Ta_2_O_5_@C/S, and TiN@C/S. (**b**) Galvanostatic charge–discharge curves of different cathodes at 0.5C. (**c**) rate performances of various sulfur electrodes. (**d**) Cycling performance of TiN@C/S/Ta_2_O_5_, Ta_2_O_5_@C/S, TiN@C/S and C/S cathodes at 0.5 C. (**e**) Cycling performances for TiN@C/Ta_2_O_5_/S pouch cells over 50 cycles at 0.2 C charge/discharge rate with a 5.3 mg cm^−2^.

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
