# Peer review of "Synergistic Adsorption-Catalytic Sites TiN/Ta2O5 with Multidimensional Carbon Structure to Enable High-Performance Li-S Batteries"

_nanomaterials, 2021, doi:10.3390/nano11112882_

Round 1
Reviewer 1 Report
This manuscript deals with the preparation of of electrodes to improve the performance of Li-sulfur batteries. a subject of high impact in the field of advanced batteries.
However, the manuscript in its actual form requires a revision before I could recommended for publication in Nanomaterials.
A first concern, probably a typing error, is the title of the manuscript, that it does not match with that provided by the editors when they invited me to review this work (Synergistic adsorption-catalytic sites TiN/Ta2O5 with multidimensional carbon structure to enable high-performance Li-S batteries)
An important point that requires clarification is the idea beyond the use of three different carbon materials to prepare the electrodes. Super P, CNT and graphene have different morphology and physical properties (such as electrical conductivities, porosity, etc.) but the authors did not discuss the rationale behind the use of the three carbonous material in their formulation. It is claim that "the multidimensional carbon structure facilitates the infiltration of electrolytes and the motion of ions and electrons throughout the framework" but the nanopore structure of the framework and the specific surface area has not been analyzed. In the SEM images only the macroporosity is visualized, while the meso and nanoporosity that determine the area available for the reaction is not well described.
I wonder what would be the performance of the electrodes if only one of the carbon material is used separately?. The authors should discuss this point that could have also an impact on the cost of the battery.
The authors use a set of characterization techniques in their study, most of them are routine when dealing with battery electrodes but I do not understand the relevance of the thermogravimetric analysis.
The authors quoted that "the cathode exhibit superior cycle stability with a decent capacity retention". What does "decent" means in this context?. Dr. Fan's group have published other works dealing with electrodes for Li-S batteries and many articles have been published by other authors during the last decade. So, it is mandatory that the authors compare their results of capacity retention with those already reported in the literature.
Minor comments:
Page 2, line 74: replace "mixed solution of TiN@C" by "mixed dispersion of TiN@C".
Page 3, line 109: replace "homogeneous solution" by "homogeneous dispersion" or "homogeneous ink"
What's the origin of the periodic discontinuities observed in Fig, 7 (c) where the specific capacity shows jumps for 5,10,15 and 20 cycles?
In summary, I would suggest the authors to consider these points in a revised version of the manuscript.
Reviewer 2 Report
This manuscript deals with the preparation of the TiN@C/S/Ta2O5 sulfur electrode containing about 90 wt% sulfur. The novelty of the material and the mechanism for the adsorption-catalytic effect are clear in the manuscript. However, the manuscript need more systematical analysis and majorly to revise for the publication in Nanomaterials.
Reviewer’ comments:
- In Figure 4(c-g) and the corresponding description, it is hard to say that Ti is uniformly distributed on the composite, TiN@C/S/Ta2O5. Please check and explain it.
- In Figure 6(a), there is no characteristic peak from the LiPSs conversion if
compared with the reported literatures. Does TiN or Ta2O5 really have any
catalytic effect for LiPS conversion? Please explain this.
- Both of CNT and graphene can contribute to LiPSs adsorption and conversion. More detail data are needed to illustrate the contribution of CNT and graphene, respectively.
- In Figure 5(d-e), the peaks of Ta 4f and O 1s move to higher binding energy after the adsorption. It should be explained why.
- In the PDF file, the title is “Scheme 2. O5 with multidimensional carbon
structure to enable high-performance Li-S batteries”. Is it correct?
Reviewer 3 Report
This article demonstrate the concept of synergistic adsorptive–catalytic sites by utilizing TiN@C/S/Ta2O5 sulfur electrode and the superior battery performance. Although, the paper focused on the multidimensional carbon structure facilitates the infiltration of electrolytes and the motion of ions and electrons throughout the framework, the details of carbon material are not mentioned in the manuscript.
- title in pdf file is not correct should be revised.
- “multidimensional carbon structure” is too ambiguous and not clear.
- In experimental section the details of Super P, CNT, Graphene is not described, such as synthetic method and basic property.
- In experimental section, size of electrode is not described.
- This research should describe the inspiration and effect of using the Super P, CNT, and Graphene for the fabrication of the TiN@C/S/Ta2O5 composite. They explained only the part of TiN and Ta2O5; thus, it does not clarify to understand that why this composite combines the carbon materials?
- Figure 7c is not mentioned in the part of results and discussion.
- The presence of Ta2O5 on the external surface and TiN@C as host should be experimentally confirmed, by analytical techniques, such as XPS or EDS.
Reviewer 4 Report
This work reports the preparation of TiN@C/S/Ta2O5 composites with high sulfur fraction via the co-precipitation method. This composite has demonstrated remarkable cyclability and rate capability, even in pouch-cell tests. The work is correctly planned, in a systematic way and the results are scientifically reasoned. The original character and the reported results are interesting for the journal Nanomaterials. However, before publication, authors should take into account the following considerations in order to increase the quality of the manuscript:
(1) The typo in the title must be corrected in the manuscript.
(2) On page 1 the authors indicate: “Catalysts such as transition metal-free polar materials [4-5], transition metal compounds [6-9], and metals [10] can not only capture LiPSs to decrease their dissolution and diffusion in the electrolyte but also boost the conversion between LiPSs and Li2S2/Li2S. " MOFs have also been reported as catalyst materials for this purpose. The authors could add examples published in this same journal [10.3390/nano10030424].
(3) In the Materials section, please indicate the type of graphene and CNT (supplier, fundamental properties, ...).
(4) In the Experimental section, please indicate if the current density (mA/g) and the specific capacities (mAh/g) refer to the mass of sulfur in the electrode.
(5) Authors should present and discuss XPS analysis of sulfur to demonstrate binding to TiN and Ta2O5.
(6) In Figure 7c, indicate the rates used in the experiment.
(7) In the discussion of Figures 7d and 7e, please calculate and discuss the capacity loss per cycle (% per cycle) in each of the experiments shown in the graphs.
Author Response
Response to Reviewer 4 Comments
Dear Reviewer,
Thank you for your useful comments and suggestions on the language and structure of our manuscript. We have modified the manuscript accordingly, and detailed corrections are listed below point by point:
Point 1: The typo in the title must be corrected in the manuscript
Response 1:The title of the manuscript is a typing error.
Point 2: On page 1 the authors indicate: “Catalysts such as transition metal-free polar materials [4-5], transition metal compounds [6-9], and metals [10] can not only capture LiPSs to decrease their dissolution and diffusion in the electrolyte but also boost the conversion between LiPSs and Li2S2/Li2S. " MOFs have also been reported as catalyst materials for this purpose. The authors could add examples published in this same journal [10.3390/nano10030424].
Response 2: I mainly focus on the catalytic effect of transition metal oxides on polysulfides
Point 3: In the Materials section, please indicate the type of graphene and CNT (supplier, fundamental properties, ...).
Response 3: The pore structures of the multidimensional carbon structure were studied by N2 adsorption-desorption analysis. The BET specified surface area of the multidimensional carbon structure was 352 m2 g-1. A dual distribution of micropore and mesopore were observed on the multidimensional carbon structure (Fig 2e-f). The electronic conductivity of was 4.38×103 S m-1.
Point 4: In the Experimental section, please indicate if the current density (mA/g) and the specific capacities (mAh/g) refer to the mass of sulfur in the electrode.
Response 4: In this experiment, the electrolyte was 1M LiTFSI in DOl/DME (1:1 v/v) containing LiNO3 as an additive (1wt%), The E/S ratio in the coin cells with areal sulfur loading (1.5mg cm-2) was controlled to be 10 μL mg-1. The pouch cells have average areal sulfur loading of 5.3 mg cm-2 and a decreased electrolyte/sulfur ratio of 3.3 μL mg-1.
Point 5: Authors should present and discuss XPS analysis of sulfur to demonstrate binding to TiN and Ta2O5.
Response 5: It is different to distinguish between TiN and Ta2O5. Because, we make use of the mixture of TiN and Ta2O5 to carry on adsorption tests. So, X-ray photoelectron spectroscopy (XPS) technique was put into use to understand the chemical interaction of catalysts before and after absorption of Li2S6 respectively.
Point 6: In Figure 7c, indicate the rates used in the experiment.
Response 6: The rate performances are displayed in Fig, 7 (c).
Point 7: In the discussion of Figures 7d and 7e, please calculate and discuss the capacity loss per cycle (% per cycle) in each of the experiments shown in the graphs.
Response 7: the coin Li-S cells with TiN@C/S/Ta2O5 cathode exhibit superior cycle stability with a decent capacity retention of 56.1% over 300 cycles and a low capacity fading rate of 0.192% per cycle at 0.5C
Round 2
Reviewer 1 Report
Authors have just did cosmetic changes in the paper and they have not addressed the points I raised. So, in my opinion the paper need further revision before I could recomend publicationAuthor Response
Thank you for your useful comments and suggestions on the language and structure of our manuscript. We have modified the manuscript accordingly, and detailed corrections are listed below point by point:

Reviewer 2 Report
Please provide more experimental data to show the advantage of TiN@C/S/Ta2O5 over the bare carbon structure.
Author Response
Thank you for your useful comments and suggestions on the language and structure of our manuscript. We have modified the manuscript accordingly, and detailed corrections are listed below point by point:

Reviewer 3 Report
the manuscuript was largely improved and worth to publish as it is